# The Diagnosis of Malignant Pleural Effusion Using Tumor-Marker Combinations: A Cost-Effectiveness Analysis Based on a Stacking Model

**DOI:** 10.3390/diagnostics13193136

**Published:** 2023-10-05

**Authors:** Jingyuan Wang, Jiangjie Zhou, Hanyu Wu, Yangyu Chen, Baosheng Liang

**Affiliations:** 1Department of Biostatistics, School of Public Health, Peking University, Beijing 100191, China; wjy123@bjmu.edu.cn (J.W.); actionsafe@pku.edu.cn (J.Z.); 1903119428w@gmail.com (H.W.); 2Department of Respiration and Critical Care Medicine, Beijing Chaoyang Hospital, Beijing 100020, China; panchenruyi@vip.sina.com

**Keywords:** malignant pleural effusion, cost-effectiveness analysis, tumor marker, stacked ensemble

## Abstract

Purpose: By incorporating the cost of multiple tumor-marker tests, this work aims to comprehensively evaluate the financial burden of patients and the accuracy of machine learning models in diagnosing malignant pleural effusion (MPE) using tumor-marker combinations. Methods: Carcinoembryonic antigen (CEA), carbohydrate antigen (CA)19-9, CA125, and CA15-3 were collected from pleural effusion (PE) and peripheral blood (PB) of 319 patients with pleural effusion. A stacked ensemble (stacking) model based on five machine learning models was utilized to evaluate the diagnostic accuracy of tumor markers. We evaluated the discriminatory accuracy of various tumor-marker combinations using the area under the curve (AUC), sensitivity, and specificity. To evaluate the cost-effectiveness of different tumor-marker combinations, a comprehensive score (C-score) with a tuning parameter *w* was proposed. Results: In most scenarios, the stacking model outperformed the five individual machine learning models in terms of AUC. Among the eight tumor markers, the CEA in PE (PE.CEA) showed the best AUC of 0.902. Among all tumor-marker combinations, the PE.CA19-9 + PE.CA15-3 + PE.CEA + PB.CEA combination (C9 combination) achieved the highest AUC of 0.946. When *w* puts more weight on the cost, the highest C-score was achieved with the single PE.CEA marker. As *w* puts over 0.8 weight on AUC, the C-score favored diagnostic models with more expensive tumor-marker combinations. Specifically, when *w* was set to 0.99, the C9 combination achieved the best C-score. Conclusion: The stacking diagnostic model using PE.CEA is a relatively accurate and affordable choice in diagnosing MPE for patients without medical insurance or in a low economic level. The stacking model using the combination PE.CA19-9 + PE.CA15-3 + PE.CEA + PB.CEA is the most accurate diagnostic model and the best choice for patients without an economic burden. From a cost-effectiveness perspective, the stacking diagnostic model with PE.CA19-9 + PE.CA15-3 + PE.CEA combination is particularly recommended, as it gains the best trade-off between the low cost and high effectiveness.

## 1. Introduction

Malignant pleural effusion (MPE) is the abnormal accumulation of fluid and malignant cells or tumor issues in the pleural space, which is mainly caused by primary cancers such as breast cancer, lung cancer, lymphomas, or secondary cancers that have metastasized to the pleura [1,2]. Unlike benign pleural effusion (BPE), which often results from non-malignant diseases like congestive heart failure and pleural inflammation, the presence of MPE signals adverse clinical outcomes. MPE is associated with high mortality and a poor prognosis, with its median survival time ranging from 4 to 7 months [3]. Therefore, an early diagnosis of the benign or malignant pleural effusion is critical to the intervention and treatment accordingly [4,5].

Traditionally, pleural effusion cytology was the simplest diagnostic method of MPE, but its accuracy ranged widely from 62% to 90% [6,7]. Utilizing the strong association between tumor markers and malignant tumors, the analysis of tumor-marker concentrations has been recognized as a fast and less invasive method to diagnose MPE. Clinically, tumor markers such as carcinoembryonic antigen (CEA), carbohydrate antigen (CA)19-9, CA125, and CA15-3 have been frequently employed, and show satisfying diagnostic accuracy [7,8,9,10,11]. Therefore, using cut-off values of tumor-marker concentrations to diagnose MPE has been popular in past years [6,7,10,11]. To handle the complex associations between tumor markers, machine learning methods, such as logistic regression, support vector machine (SVM), random forest, etc., have been widely used in this field and show good performance [12,13,14]. However, faced with so many algorithms, which one to choose in a real application becomes a challenging issue. The stacked ensemble, or stacking, is a heterogeneous ensemble method that learns from the outputs of several distinct algorithms, providing a combined output and thus sidestepping the model selection dilemma. What is more, it has proved to perform at least comparably with the best individual algorithm included in the ensemble [15]. Since its introduction in the early 1990s [16], the method has been utilized in many fields because of its potential to enhance the diagnostic accuracy, as well as the convenience of avoiding the model selection procedure [17,18,19,20,21,22]. 

Despite the diagnostic accuracy of tumor markers, some studies discovered that the discriminating ability of single tumor markers was not high enough to make a precise MPE diagnosis, indicating the need for multi-marker combinations [13,14]. But the more tumor markers tested, the more costs and medical resources are consumed. Most studies focused only on the diagnostic accuracy, but ignored the financial burden on patients and the medical system [12,13,23,24,25,26]. Moreover, many studies have found minimal differences in AUC between single markers and multi-marker combinations. Notably, the diagnostic accuracy of multi-marker combinations does not always rise with the inclusion of more tumor markers [10,11,14]. Therefore, the cost-effectiveness issue, especially the necessity of including many tumor markers in diagnosing MPE, needs to be evaluated to give comprehensive recommendations on tumor-marker selection.

The objective of this study is to balance the diagnostic accuracy and economic burden in diagnosing MPE using tumor-marker combinations. We first utilized a stacking algorithm to enhance the diagnostic accuracy. Then, we proposed a cost-effectiveness analysis framework incorporating both the diagnostic accuracy and the price of tumor markers’ test. Finally, recommendations on choosing diagnostic models of MPE based on the results of the cost-effectiveness analysis were tailored for patients in different economic levels.

## 2. Materials and Methods

### 2.1. Study Population and Diagnostic Criteria 

A total of 319 patients with pleural effusion were included retrospectively from January 2018 to June 2020, where 174 patients were admitted to the Department of Respiratory and Critical Care Medicine, Beijing Chaoyang Hospital, Capital Medical University, and 145 were admitted to the Central Hospital of Wuhan. Patients were categorized to MPE and BPE groups based on pathological results, specifically the presence of malignant tumor cells in pleural effusion or biopsy specimens. 

BPE etiologies include tuberculosis, pneumonia, congestive heart failure, and others. The diagnostic criteria for tuberculous pleural effusion are positive pleural effusion, sputum, pleural biopsy specimens stained with a Ziehl–Neelsen/Lowenstein–Jensen culture, or tuberculosis granuloma found in parietal pleural biopsy; pleural effusion associated with bacterial pneumonia, lung abscess, and bronchiectasis infection was determined if the effusion disappeared after anti-infective treatment; heart failure associated pleural effusion was judged with cardiac color Doppler ultrasound, imaging examinations, and other relevant diagnostic criteria. Pleural effusion with undetermined etiology was excluded from this study.

### 2.2. Specimen Collection and Measurement 

Pleural effusion (PE) and peripheral blood (PB) samples were collected from patients prior to any treatment. Samples from all patients were centrifuged on the same day (below 4 °C, 1500 rpm, 10 min) and the supernatant was taken and frozen at −80 °C. The concentration of 4 tumor markers including CEA, CA125, CA15-3, and CA19-9 was detected from the PE and PB, respectively. The chemiluminescence method was used to detect CEA (Abbott Ireland Diagnostics Division, Sligo, Ireland), CA125, CA15-3, and CA19-9 (Abott Laboratories, Malvern, PA, USA) tumor markers.

### 2.3. The Construction of Stacking Model for Discriminating MPE

We employed five machine learning models, i.e., logistic regression, random forest, Naïve Bayes, SVM, and XGBoost, to distinguish MPE from BPE. A stacked ensemble (stacking) model built with these five learners was further utilized to enhance the diagnostic accuracy and avoid model selection problems. Logistic regression was taken as the meta-learner to stack the output of the five learners and give the prediction of the stacking model.

Baseline characteristics, including age and gender, were incorporated in the model alongside tumor-marker concentrations. To avoid over-fitting, 3-fold cross-validation was performed. Hyperparameters of machine learning methods were optimized using internal cross-validation in the training sets. The area under the curve (AUC), sensitivity, and specificity of the validation cohort were calculated to evaluate the model performance.

### 2.4. Cost-Effectiveness Analysis of Tumor-Marker Combinations

As previously mentioned, eight tumor markers were available from both PE and PB samples. There were 255 different combinations (2^8^-1) of the eight tumor markers. Together with important clinical factors, the AUC, sensitivity, and specificity of all these 255 combinations were calculated to evaluate the effectiveness.

For the cost, the price of performing different tumor-marker tests was collected from the public information on the website of the Healthcare Security Administration of Beijing and Hubei province, respectively [27,28]. The price is set by the government and remains fixed over the years, so there is no need to adjust for the inflation and variations. The cost of each tumor-marker combination was calculated as the summation of the cost for each single tumor-marker element.

To align with AUC values, we regularized the cost of tumor-marker combinations to the [0,1] interval using minimum and maximum values. A comprehensive score (C-score) was proposed to comprehensively evaluate the diagnostic accuracy and the cost, which was defined as
C-score=w×AUC+(1−w)×(1−regularized(cost)),
where the regularized ({$x_{1},\ldots ,x_{n}\})=\left\{\frac{x_{i}-\min\left(\left\{x_{1},\ldots ,x_{n}\right\}\right)}{\max\left(\left\{x_{1},\ldots ,x_{n}\right\}\right)-\min\left(\left\{x_{1},\ldots ,x_{n}\right\}\right)}:i=1,\ldots ,n\right\}$; *w* indicates the weight allocated to AUC. The larger the value of the C-score, the better cost-effectiveness achieved. This C-score was simple but easy for interpretation and calculation.

### 2.5. Statistical Analysis 

Concentrations of tumor markers and patients’ clinical factors were calculated as the mean ± standard deviation or proportion. To compare the difference between MPE and BPE groups, a Mann–Whitney–Wilcoxon test was used for continuous variables, and a Pearson χ^2^ test or Fisher exact probability test was used for categorical variables, which both have no assumptions on the distribution of data. The hypothesis tests were performed using package stats (version 4.2.0) in R software (version 4.2.0); the diagnostic models and statistical plots were generated with packages scikit-learn (version 0.24.1), scipy (version 1.10.1), xgboost (version 1.4.2), matplotlib (version 3.7.1), and so on in Python (version 3.6.13). A *p*-value of the two-sided hypothesis test smaller than 0.05 was considered statistically significant.

## 3. Results

### 3.1. Clinical Characteristics

The characteristics of the included patients are summarized in Table 1. In this study, 111 patients with MPE and 208 patients with BPE were included. For the etiology of BPE patients, 139 were tuberculous, 27 were pneumonitis, 30 were congestive heart failure, and 12 were others. In addition to tumor markers, demographic data such as age, gender, and region of patients were collected. The analysis indicated a significant association between MPE and factors such as age and gender, while regional differences were not statistically significant, which was not included in the following model construction. The concentrations of CEA, CA19-9, CA125, and CA15-3 from PE and PB in the MPE group were all significantly higher than those from the BPE group (all *p*-values < 0.001).

### 3.2. The Cost of Different Tumor-Marker Combinations

Prices for various tumor-marker tests were separately collected for both Beijing and Wuhan. There was no difference between the price of CA125, CA15-3, and CA19-9. The price of testing tumor markers from PB and PE was also the same. Therefore, although there were 255 (2^8^ − 1) different combinations of tumor markers, only twenty distinct cost values were generated.

### 3.3. The Diagnostic Performance of the Stacking Model

The AUC of the five machine learning models and the stacking model is displayed in Figure 1. Given that there were only twenty distinct cost values, only the model with the highest AUC is displayed for each value (denoted as C1-C20; see Table 2). It is explicit that the AUC of the stacking model was generally the best (at least no worse than the others) among all tumor-marker models.

### 3.4. The Diagnostic Performance of Different Tumor-Marker Combinations

The AUC, sensitivity, and specificity of the diagnostic models using C1-C20 tumor-marker combinations are listed in Table 2. The AUC ranged from 0.871 to 0.946; the sensitivity ranged from 0.595 to 0.829; and the specificity ranged from 0.913 to 0.966. Among single tumor markers, PE.CEA achieved the best AUC of 0.902, a sensitivity of 0.748, and specificity of 0.937. As depicted in Figure 1, combinations C2, C5, C8, C11, C14, and C17 emerged like outliers, which presented relatively low AUC compared to other combinations (see Figure 2). The common feature of these outlier combinations was they did not include the best single tumor marker PE.CEA. Among all tumor-marker combinations, the combination with the highest AUC was C9 (i.e., PE.CA19-9 + PE.CA15-3 + PE.CEA + PB.CEA), which reached an AUC of 0.946, a sensitivity of 0.748, and specificity of 0.923.

### 3.5. Cost-Effectiveness Analysis

Figure 2 presents a scatter plot illustrating the relationship between AUC and the price of different tumor-marker combinations. After removing outliers, a distinct trend emerged: single or two-marker combinations (C1-C4) had suboptimal AUC values. The AUC increased with combinations of three and four tumor markers (C6-C10), but stagnated when five or more markers were included (C12-C20).

Based on the median values of AUC and cost, the 20 combinations were segmented into four subgroups: (1) combinations like C7, C9, and C10 with relatively high AUC and low cost; (2) combinations like C1-C6 and C8 with lower AUC and low cost; (3) combinations characterized by high AUC and high cost; and (4) combinations exhibiting low AUC and high cost.

Figure 3 displays the C-score of C1-C20 in some representative *w* values. For each value of *w*, each dot represents a tumor-marker combination. When *w* was no more than 0.80, the combination with the highest C-score was always the cheapest combination C1, namely the single marker PE.CEA, and the combination with the lowest C-score was always the most expensive one, C20, i.e., the all-markers combination. The color of dots also confirmed that combinations with lower cost usually achieved higher C-scores. But when *w* increased to 0.90 or higher, which meant much more importance was put on diagnostic accuracy rather than cost, more expensive marker combinations gradually achieved higher C-scores, and the combinations with the highest and lowest C-score were not C1 and C20 anymore. In specific, the combinations with the highest and lowest C-score were C4 and C17 when *w* equaled 0.90; C7 and C17 when *w* equaled 0.95; and C9 and C2 when *w* equaled 0.99. The changes in C-scores of C1 and C20 across different *ws* are typical examples, which are represented with the blue and green dashed lines. When *w* increased, the C-score of C20 increased rapidly and finally exceeded C1 (C-score: 0.928 vs. 0.903 when *w* equaled 0.99).

## 4. Discussion

It has been a popular way to collect multiple tumor markers and combine them into a model in diagnosing MPE. When multiple markers with high correlation and interaction were included in a model, many machine learning methods performed well [23,24,29,30]. However, when dealing with real-world data, there are numerous methods. Selecting the right model is critical. The stacking method provides a convenient way to avoid the model selection procedure by synthesizing the outputs of different models [25]. By incorporating the strength of different models, it also has the potential to enhance the diagnostic performance. Our analysis revealed that the stacking model effectively improved diagnostic accuracy for the majority of tumor-marker combinations. Higher diagnostic accuracy will benefit the early diagnosis of malignant pleural effusion, which could result in better outcomes with early and precise treatment [15].

Of the eight single tumor markers examined, the single marker model with PE.CEA yielded the highest AUC at 0.902. In addition, in the multi-marker situation, each tumor-marker combination without PE.CEA was an explicit outlier with a lower AUC compared with other combinations (see Figure 1), indicating the strong relevance between PE.CEA and MPE. This is similar to other studies [1,13,14], which also found that CEA performed best compared with other tumor markers. Combining multiple tumor markers has the potential to further enhance diagnostic accuracy [31,32]. Zhang [13] et al. also concluded that most tumor markers had insufficient diagnostic accuracy to confirm or exclude MPE when used alone. In this study, all combinations of the eight tumor markers were tested, and the AUC was increased compared to single markers. The highest AUC of 0.946 was achieved with the C9 combination (PE.CA19-9 + PE.CA15-3 + PE.CEA + PB.CEA), which is higher than other relevant studies [14,24,25]. However, the AUC did not increase further when more tumor markers were included (see Figure 1). A possible reason for this may be the strong correlation of different tumor markers, which means that the combination with less markers can include all useful information in diagnosing MPE. As a result, when incorporating more markers on C9, the multicollinearity and model complexity issue resulted in a lower model performance. It is explicit in Table 2 that the sensitivity ranging from 0.595 to 0.826 was lower and more variable than specificity (ranging from 0.913 to 0.966). This is also consistent with many previous studies [1,4,14]. Therefore, when utilizing tumor markers to diagnose MPE, patients with positive predictions are very likely to develop MPE. But patients with negative predictions should receive more attention, and more invasive testing, such as cytology sampling or pleural biopsies, should be considered if necessary.

As a serious syndrome associated with a poor prognosis, the diagnosing and treatment of MPE have attracted much attention. There were also many studies investigating the cost-effectiveness of MPE treatment or cytopathologic examination techniques [5,15,33]. For example, Aaron et al. [15] compared the Pleurx Catheter and Talc Pleurodesis treatment method for MPE by obtaining the cost and probability of treatment success, and concluded that the Pleurx Catheter was more cost-effective when life expectancy was 6 weeks or less. Varun et al. [5] used a decision analysis and found that tunneled pleural catheter use was the preferred treatment for patients with malignant pleural effusion and limited survival, and bedside pleurodesis was the most cost-effective treatment for patients with more prolonged expected survival. In contrast, the cost-effectiveness of conducting MPE prediction with tumor markers was less investigated.

Incorporating additional tumor markers in the diagnosis of MPE increases the associated costs. In this study, the price of detecting different tumor markers was collected and used for the cost-effectiveness analysis. Among different tumor-marker combinations, the AUC increased when two or three markers were included compared with a single marker. But when five or more markers were included, the AUC failed to increase. Therefore, there were several tumor-marker combinations that exhibited relatively low cost and high AUC, such as C7, C9, and C10 (see Figure 2). A weighted average of AUC and negative regularized cost, namely the C-score, was also proposed as an indicator of cost-effectiveness. For most *w* values, combinations with less cost had a higher C-score. Specifically, the cheapest combination C1, which only included the single marker PE.CEA, had the highest C-score when *w* ranged from 0 to 0.8 (Figure 3). The reason behind that was that the difference between the cost of tumor-marker combinations was much larger than AUC values. The C-score favored more expensive combinations when the weight *w* put on AUC got high, i.e., the diagnostic accuracy was valued more than cost. When *w* equaled 0.95, the highest C-score was achieved with C7, which indicated its cost-effectiveness again. When *w* equaled 0.99, meaning that almost all importance was put on diagnostic accuracy, the combination C9, which achieved the best AUC, also had the highest C-score.

As a result, after taking the financial burden into consideration, different recommendations could be given to patients in different economic levels on diagnosing MPE with tumor-marker combinations. For patients without a sufficient budget on medical expenses, the marker PE.CEA is the best choice with the lowest cost and not bad AUC of 0.902. For most patients that aim to balance the cost and effectiveness, C7 is a good choice, which balances the cost and effectiveness well. For patients in an excellent economic level, the tumor-marker combination with the highest AUC, namely C9, is more suitable.

This study also had several limitations. Firstly, there are many types of tumor markers associated with MPE, and some markers like the cytokeratin fragment (CYFRA) 21-1 and cluster of differentiation 66 (CD66) antigen, which showed good MPE diagnostic performance in previous studies, were not considered in this study [12,26]. Secondly, as a cost-effectiveness analysis, we only considered the money spent on performing different tumor-marker tests. Other aspects such as the patient experiences on sampling PE and PB could also be quantified and considered in the future. In addition, the cost collected in this study was the price in tertiary hospitals in Beijing and Wuhan, but the price of secondary hospitals and a primary care center was different, which limits the generalization of the conclusion. Lastly, the results and discussion in this article, as well as the recommended solutions, are purely from the cost-effectiveness perspective for a single diagnosis. However, from a medical perspective, life is precious. In the face of major health problems, sometimes the cost is even not an important factor, and sometimes it may require overall consideration for the cost-effectiveness analysis, including costs in both diagnostic part and treatment part. In practice, strict clinical guidelines and medical ethical standards are required to determine which diagnostic model scheme is applicable to specific patients. In future studies, more tumor markers will be tested and incorporated into our cost-effectiveness study; more clinical baseline factors would be collected. To enhance the generalizability of our findings, patients from other centers and hospitals will also be included, and some of them could serve as external validation. As for the cost, more precise and individualized data, such as the insurance and income of patients, will be collected under appropriate ethical instruction and approval.

## 5. Conclusions

This study employed the stacking model to improve the diagnostic accuracy of machine learning models for MPE using tumor markers. Among the eight tumor markers, the most accurate one was PE.CEA. The most accurate tumor-marker combination was the C9 combination (PE.CA19-9 + PE.CA15-3 + PE.CEA + PB.CEA). By factoring in the costs associated with tumor-marker tests, this study assessed the cost-effectiveness of an MPE diagnosis using various tumor-marker combinations. The findings provide guidance on tumor-marker selection tailored to patients’ economic circumstances. For economically constrained patients, extensive testing is unnecessary. The marker PE.CEA emerges as the optimal choice due to its low cost and an AUC of 0.902. For most patients in favor of cost-effectiveness, C7 is a good choice that balanced the cost and benefit well. For affluent patients, greater emphasis can be placed on diagnostic accuracy, C9 is recommended, which had the highest AUC of 0.946.

## Figures and Tables

**Figure 1 diagnostics-13-03136-f001:**
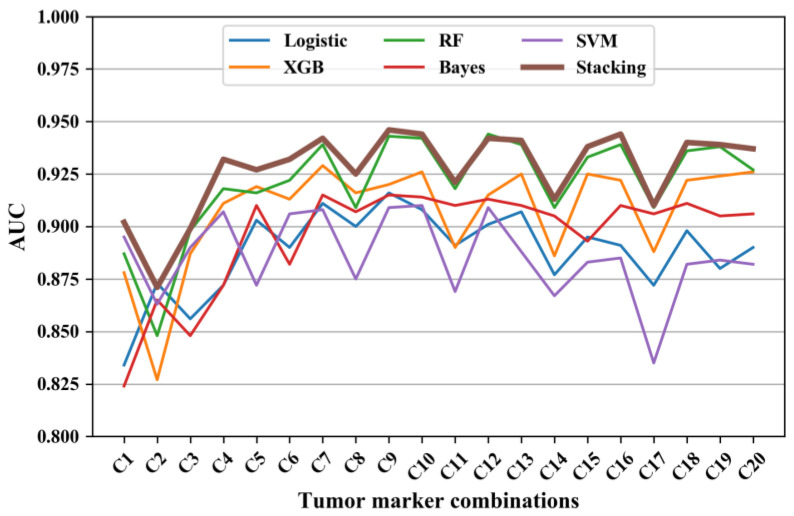
The AUC of diagnosing models using different tumor-marker combinations. Note: Logistic represents logistic regression, XGB represents XGBoost model, RF represents random forest, Bayes represents Naïve Bayes, SVM represents SVM model, and Stacking represents the stacking model.

**Figure 2 diagnostics-13-03136-f002:**
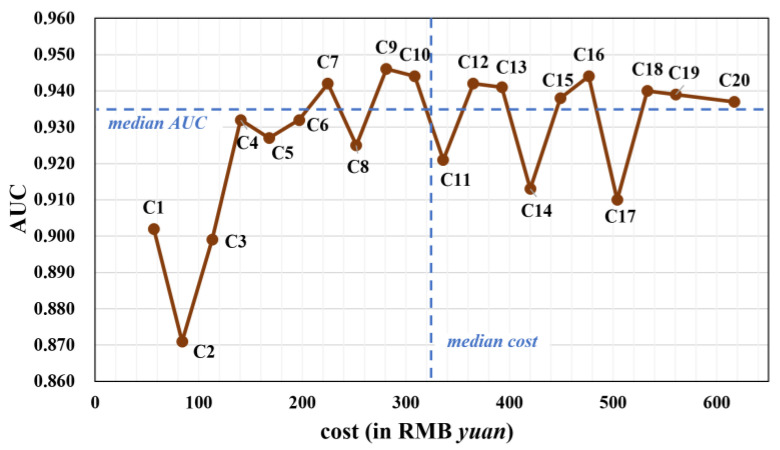
The AUC and the cost of different tumor-marker combinations using stacking model (in gray color). The median cost (322.25) and median AUC (0.935) are plotted as dashed blue lines.

**Figure 3 diagnostics-13-03136-f003:**
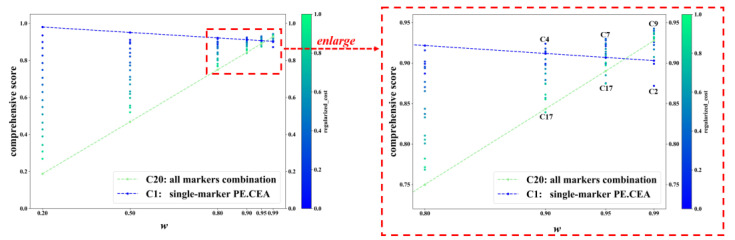
The C-score of different tumor-marker combinations on different *w*. Note: Each dot represents a tumor-marker combination. Regularized_cost is the min–max-regularized value of the cost. *w* is the weight of AUC in calculating comprehensive score. C-score = *w* × AUC + (1−*w*) × (1− regularized_cost). C1-C20 represent different tumor-marker combinations (see Table 2).

**Table 1 diagnostics-13-03136-t001:** Clinical characteristics of patients and concentration of tumor markers.

Variable	MPE (N = 111)	BPE (N = 208)	Total (N = 319)	*p*-Value
Age (years)	60.94 (11.96)	49.78 (19.02)	53.67 (17.70)	<0.001
Gender				0.002
Male	59	148	207	
Female	52	60	112	
Region				0.160
Beijing	67	107	174	
Wuhan	44	101	145	
Concentrations of tumor markers				
PE.CEA (ng/mL)	967.15 (2307.31)	16.80 (105.81)	347.49 (1433.20)	<0.001
PE.CA19-9 (U/mL)	2000.31 (4200.90)	5.74 (12.94)	699.77 (2647.65)	<0.001
PE.CA125 (U/mL)	1807.16 (2473.70)	715.22 (796.83)	1095.17 (1673.73)	<0.001
PE.CA15-3 (U/mL)	99.97 (199.45)	7.48 (7.80)	39.66 (125.49)	<0.001
PB.CEA (ng/mL)	98.16 (254.14)	2.12 (1.74)	35.54 (156.34)	<0.001
PB.CA19-9 (U/mL)	628.56 (2312.09)	13.88 (40.28)	227.76 (1391.48)	<0.001
PB.CA125 (U/mL)	257.72 (502.10)	155.21 (147.49)	190.88 (322.11)	<0.001
PB.CA15-3 (U/mL)	38.79 (47.97)	12.01 (9.21)	21.33 (31.85)	<0.001

Note: Continuous variables are displayed using the mean and standard deviation (in parentheses) in each group; discrete variables are displayed using the number in each group.

**Table 2 diagnostics-13-03136-t002:** The AUC of stacking models using different tumor-marker combinations.

Tumor-Marker Combination	Cost	AUC	SEN	SPE
C1: PE.CEA	56.5	0.902	0.748	0.937
C2: PE.CA15-3	84	0.871	0.595	0.962
C3: PE.CEA + PB.CEA	113	0.899	0.766	0.913
C4: PE.CA19-9 + PE.CEA	140.5	0.932	0.757	0.933
C5: PE.CA19-9 + PE.CA15-3	168	0.927	0.721	0.957
C6: PE.CA19-9 + PE.CEA+PB.CEA	197	0.932	0.748	0.923
C7: PE.CA19-9 + PE.CA15-3+PE.CEA	224.5	0.942	0.775	0.942
C8: PE.CA19-9 + PE.CA15-3+PB.CA15-3	252	0.925	0.739	0.952
C9: PE.CA19-9 + PE.CA15-3+PE.CEA+PB.CEA	281	0.946	0.748	0.928
C10: PE.CA19-9 + PE.CA15-3+PE.CEA+PB.CA125	308.5	0.944	0.757	0.957
C11: PE.CA19-9 + PE.CA15-3+PB.CA19-9+PB.CA125	336	0.921	0.712	0.947
C12: PE.CA19-9 + PE.CA15-3+PE.CEA+PB.CA19-9+PB.CEA	365	0.942	0.748	0.942
C13: PE.CA19-9 + PE.CA125+PE.CA15-3+PE.CEA+PB.CA125	392.5	0.941	0.829	0.966
C14: PE.CA19-9 + PE.CA15-3+PB.CA19-9+PB.CA125+PB.CA15-3	420	0.913	0.730	0.961
C15: PE.CA19-9 + PE.CA125+PE.CEA+PB.CA125+PB.CA15-3+PB.CEA	449	0.938	0.775	0.952
C16: PE.CA19-9 + PE.CA125+PE.CA15-3+PE.CEA+PB.CA19-9+PB.CA125	476.5	0.944	0.793	0.957
C17: PE.CA19-9 + PE.CA125+PE.CA15-3+PB.CA19-9+PB.CA125+PB.CA15-3	504	0.910	0.712	0.947
C18: PE.CA19-9 + PE.CA125+PE.CA15-3+PE.CEA+PB.CA19-9+PB.CA125+PB.CEA	533	0.940	0.802	0.957
C19: PE.CA19-9+PE.125+PE.CA15-3+PE.CEA+PB.199+PB.125+PB.CA15-3	560.5	0.939	0.820	0.942
C20: PE.CA19-9+PE.CA125+PE.CA15-3+PE.CEA+PB.199+PB.CA125+PB.CA15-3+PB.CEA	617	0.937	0.793	0.961

Note: AUC represents area under the curve; SEN represents sensitivity; SPE represents specificity. Cost is denominated in RMB.

## Data Availability

The original data are not publicly available due to restrictions aimed to protect patient confidentiality. The data presented in this paper are available from the corresponding author upon request.

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
