# Peer review of "The Diagnosis of Malignant Pleural Effusion Using Tumor-Marker Combinations: A Cost-Effectiveness Analysis Based on a Stacking Model"

_diagnostics, 2023, doi:10.3390/diagnostics13193136_

Round 1

Reviewer 1 Report

Very interesting study, acceptable for publication.

Author Response

Please see the attached response letter. 

Reviewer 2 Report

The following points are essential for improving the quality and transparency of the paper.  • It was reported in the Discussion: "Of the eight single tumor markers examined, the single marker model with PE.CEA 225 yielded the highest AUC at 0.902.". What is the meaning of AUC at 0.902? • In terms of grammar, the English language of the paper should be improved. • The "Introduction" part of the study should be expanded, considering the research objectives, problems, and hypotheses.  • The primary output/endpoint variable(s)/measurement(s) of the study should be defined. • How was the sample size determined? This information should be explained in the Materials and Methods section.  • Which sampling (probable or non-probable, etc.) method was used in the study?  • Statistical tests for hypothesis testing and their assumptions should be specified in the study's statistical analysis in the Materials and Methods section.  • The details (version, license number, etc.) of the statistical package(s) or program(s) should be given in the section of "Data Analysis or Statistical Analysis".  • The exact P values should be added to the table(s) (e.g., p=0.25; p=0.03).  • Are the data subjected to pre-processing?  • How were extreme/outlier values in the data determined and resolved?  • Which metrics were used in the performance evaluation of the estimates of models/algorithms?  • How were the predictive models selected in this study? • Which method(s) was/were used to optimize the hyperparameters of models/algorithms? The following points are essential for improving the quality and transparency of the paper.  • It was reported in the Discussion: "Of the eight single tumor markers examined, the single marker model with PE.CEA 225 yielded the highest AUC at 0.902.". What is the meaning of AUC at 0.902? • In terms of grammar, the English language of the paper should be improved. • The "Introduction" part of the study should be expanded, considering the research objectives, problems, and hypotheses.  • The primary output/endpoint variable(s)/measurement(s) of the study should be defined. • How was the sample size determined? This information should be explained in the Materials and Methods section.  • Which sampling (probable or non-probable, etc.) method was used in the study?  • Statistical tests for hypothesis testing and their assumptions should be specified in the study's statistical analysis in the Materials and Methods section.  • The details (version, license number, etc.) of the statistical package(s) or program(s) should be given in the section of "Data Analysis or Statistical Analysis".  • The exact P values should be added to the table(s) (e.g., p=0.25; p=0.03).  • Are the data subjected to pre-processing?  • How were extreme/outlier values in the data determined and resolved?  • Which metrics were used in the performance evaluation of the estimates of models/algorithms?  • How were the predictive models selected in this study? • Which method(s) was/were used to optimize the hyperparameters of models/algorithms?

Author Response

(The authors gave the same response as above.)

Reviewer 3 Report

:Method

2.1. The description of the study population is clear, including the number of patients and the criteria used to categorize them into MPE and BPE groups. However, if it is possible, it would be beneficial to provide more information about the distribution of underlying conditions within the BPE group (e.g., how many had tuberculosis, pneumonia, etc.) for a better understanding of the control group.

2.4. Cost-effectiveness Analysis: The description of the cost-effectiveness analysis is comprehensive and well-structured. The use of real-world cost data from public sources is commendable. However, the authors should clarify whether the cost data were adjusted for inflation or any other relevant factors to ensure that the values are up-to-date. Moreover, discussing potential sources of cost variability, such as differences in healthcare systems or geographic regions, would add depth to the analysis.

Results:

3.1. The significant differences in concentrations of CEA, CA19-9, CA125, and CA15-3 between the two groups are important findings and suggest the potential diagnostic utility of these markers. Providing standard deviation for each marker in both groups would enhance the description of the data. How many times has the data measurement of each sample been repeated?

3.3. The presentation of AUC values in Figure 1 and the identification of the best-performing combination (C9) are clear and informative. However, if possible, providing specific sensitivity and specificity values for C9 and other notable combinations would offer a more complete picture of their diagnostic accuracy.

conclusion:

The authors should discuss potential limitations, such as variations in healthcare systems and costs across different regions. Is this method really affordable for all patients in all parts of the world and do all relevant medical centers have the necessary facilities?

Author Response

(The authors gave the same response as above.)

Reviewer 4 Report

While the paper titled "Diagnosis of malignant pleural effusion using tumor-marker combinations: A cost-effectiveness analysis based on stacking model" presents an interesting approach to diagnosing malignant pleural effusion (MPE) and assesses the financial implications, there are several critical comments and areas for improvement that should be considered:

·        The study appears to focus on a specific set of tumor markers (CEA, CA19-9, CA125, CA15-3) and a specific patient population (319 patients with pleural effusion). This limited scope may restrict the generalizability of the findings to a broader patient population with different tumor markers and clinical characteristics.

·        The accuracy of machine learning models heavily depends on the quality and representativeness of the data used for training and testing. The paper should address potential sources of bias and provide details on how data were collected, including any potential selection bias in the patient cohort.

·        The study does not mention external validation of the developed stacking model, which is crucial to assess its performance in new, unseen datasets. Without external validation, the model's real-world applicability remains uncertain.

·        The cost-effectiveness analysis relies on a comprehensive score (C-score) with a tuning parameter (w) that represents the weight on cost. The paper should provide more details on how these cost calculations were performed, including factors such as laboratory costs, insurance coverage, and variations in pricing.

·        The paper mentions that certain tumor-marker combinations are recommended based on cost-effectiveness, but it lacks a discussion on the clinical significance of these findings. Does the highest AUC necessarily translate into improved patient outcomes, or are there other clinical considerations to take into account?

·        The paper should provide more transparency regarding the machine learning models used, their hyperparameter tuning, and feature selection. This would allow other researchers to replicate the study and assess the robustness of the findings.

·        The paper does not discuss potential future directions or limitations of the current study. It could benefit from suggestions for further research, such as exploring additional tumor markers or incorporating clinical variables into the model.

In summary, while the paper presents a potentially valuable contribution to the field of MPE diagnosis and cost-effectiveness analysis, it would benefit from addressing the above-mentioned limitations and providing a more comprehensive and transparent analysis. Additionally, ethical considerations related to the economic status of patients should be carefully examined.

Author Response

(The authors gave the same response as above.)

Round 2

Reviewer 2 Report

Accept in present form

Accept in present form

Reviewer 4 Report

Accept